# Enhancing chronic wound healing with Thai indigenous rice variety, Kaab Dum: Exploring ER stress and senescence inhibition in HaCaT keratinocyte cell line

Witchuda Payuhakrit[1,2], Pimchanok Panpinyaporn[1], Wilunplus Khumsri[3], Gorrawit Yusakul[4], Ratsada Praphasawat[5], Nitra Nuengchamnong[6], Sarawoot Palipoch[7]*

1 Department of Pathobiology, Faculty of Science, Mahidol University, Bangkok, Thailand, 2 Pathobiology Information and Learning Center, Department of Pathobiology, Faculty of Science, Mahidol University, Bangkok, Thailand, 3 Interdisciplinary Program of Biomedical Sciences, Graduate School, Chulalongkorn University, Bangkok, Thailand, 4 School of Pharmacy, Walailak University, Nakhon Si Thammarat, Thailand, 5 Department of Pathology, School of Medicine, University of Phayao, Phayao, Thailand, 6 Science Lab Centre, Faculty of Science, Naresuan University, Phitsanulok, Thailand, 7 School of Medicine, Walailak University, Nakhon Si Thammarat, Thailand

* spalipoch@hotmail.com, sarawoot.pa@wu.ac.th

**Data Availability Statement:** All relevant data are within the paper and its Supporting Information files.

## Abstract

Kaab Dum, a prominent indigenous rice variety cultivated in the Pak Phanang Basin of Nakhon Si Thammarat, Thailand, is the focus of our study. We investigate the therapeutic potential of indigenous Kaab Dum rice extract in the context of chronic wounds. Our research encompasses an examination of the nutritional compositions and chemical profiles of Kaab Dum rice extract. Additionally, we assess how the extract affects chronic wounds in TGF-β-induced HaCaT cells. Our evaluation methods include the detection of cellular oxidative stress, the examination of endoplasmic reticulum (ER) stress, wound healing assays, analysis of cell cycle arrest and the study of cellular senescence through senescence-associated β-galactosidase (SA-β-gal) staining. Our research findings demonstrate that TGF-β induces oxidative stress in HaCaT cells, which subsequently triggers ER stress, confirmed by the expression of the PERK protein. This ER stress results in cell cycle arrest in HaCaT cells, characterized by an increase in p21 protein, a cyclin-dependent kinase inhibitor (CDKI). Ultimately, this leads to cellular senescence, as confirmed by SA-β-gal staining. Importantly, our study reveals the effectiveness of Kaab Dum rice extract in promoting wound healing in the chronic wound model. The extract reduces ER stress and senescent cells. These beneficial effects are potentially linked to the antioxidant and anti-inflammatory properties of the rice extract. The findings of our study have the potential to make significant contributions to the development of enhanced products for both the prevention and treatment of chronic wounds.

**Funding:** the Office of the Royal Development Projects Board, Thailand.

**Competing interests:** The authors have declared that no competing interests exist.

## Introduction

Rice (*Oryza sativa* L.) is an essential staple food for a significant portion of the global population. It is rich in carbohydrates, low in fat, and packed with essential nutrients like vitamins, protein, minerals and dietary fiber [1]. In southern Thailand, specifically the Pak Phanang Basin of Nakhon Si Thammarat province, numerous indigenous rice varieties flourish through extensive cultivation. These indigenous rice varieties can exhibit diverse nutritional profiles, with some being rich in specific nutrients like antioxidants, dietary fiber, vitamins or minerals, making them valuable for nutrition and health [2]. Based on data collected in the field, Kaab Dum is among the most prominent indigenous rice variety cultivated in the Pak Phanang Basin. As reported by Thailand's Rice Department, Kaab Dum rice is an indigenous rice variety with a history of cultivation spanning more than 50 years. In terms of physical appearance, the grain husks are straw-colored, while the polished rice is white. It is categorized as non-glutinous rice. Kaab Dum rice is characterized by its upright growth, with dark purple stems, leaves that are a mix of purple and green, and purple husks. The grains are short, and each rice stalk produces approximately 300–400 grains.

Chronic wounds pose a significant medical challenge, marked by inadequate healing processes and extended recovery periods, encompassing conditions such as diabetic ulcers, pressure wounds and vascular lesions [3]. Global epidemiological data emphasize the prevalence of chronic wounds, which range from 1.51 to 2.21 cases per 1,000 individuals, with a notable incidence among the elderly [4, 5]. The etiology of chronic wounds is multifaceted, involving factors like overproduction of proinflammatory cytokines, oxidative stress, cellular senescence, infections and impaired stem cell function [6, 7]. Persistent inflammation, a pivotal aspect of wound healing, impedes proper healing, resulting in chronic wounds [8]. Transforming growth factor-β (TGF-β), a crucial cytokine in wound healing, has a dual role, initially facilitating healing and potentially contributing to chronic wound pathogenesis when its activity is prolonged [9–11]. Apart from the TGF-β, senescence was also previously reported as another main factor in chronic wound association [12]. Cellular senescence, characterized by halted cell functionality and cell cycle arrest, can arise from various factors, including genotoxic stress, metabolic alterations, oxidative stress, mitochondrial dysfunction or activation of oncogenes. Emerging research indicates that cellular senescence plays a key role in chronic wound development, affecting cell reduction through cytokine production from senescent cells [12, 13].

Numerous lines of evidence support a strong correlation between cellular senescence, inflammation and chronic wound progression. Cellular senescence is a regulatory response triggered by various forms of cellular stress, including DNA damage, telomere erosion, oxidative damage and protein misfolding [14]. In chronic wounds, senescent cells accumulate, leading to a persistent inflammatory environment that impairs stem cells, affecting processes like angiogenesis, matrix remodeling, cell plasticity and growth [15]. These senescent cells maintain viability but exhibit changes in metabolic activity and develop a complex senescence-associated secretory phenotype (SASP), which can compromise tissue repair and regeneration [16]. SASP components, including cytokines, matrix remodeling proteins and growth factors can change the microenvironment to become an inflammation which is the main factor contributing to chronic wound development. Therefore, cellular senescence plays a significant role in developing chronic wounds, and strategies aimed at preventing cell senescence are emerging as promising therapeutic approaches.

This study aims to investigate the therapeutic potential of indigenous Kaab Dum rice extract in chronic wound. Specifically, it examines how Kaab Dum rice extract modulates TGF-β-induced responses in human keratinocytes (HaCaT cell line) and its potential in

mitigating cellular senescence-associated chronic wound development. Additionally, the research explores the effectiveness of Kaab Dum rice extract in stimulating wound healing through profound mechanisms, involving the attenuation of oxidative stress and endoplasmic reticulum (ER) stress.

## Materials and methods

### Rice extraction

The polished (white) and unpolished (brown) Kaab Dum rice obtained from the Rice Research Center in Nakhon Si Thammarat Province, Thailand were ground into fine powder, with each sample weighing 100 g (n = 3). Ethanol: water 18% v/v was added. The mixture was stirred, and extraction was performed using the ultrasonicate-assisted extraction technique for 90 sec at room temperature. The extract was then filtered to collect the extracted substances and the residual rice grains were extracted two more times using the same process. The substances obtained from all three extractions were combined and solvent evaporation was carried out using an air-evaporation instrument and freeze-drying.

### Analysis of nutritional compositions of Kaab Dum rice extract

The Kaab Dum rice extract underwent analysis at the Central Laboratory in Thailand. The analysis covered a range of nutrients and components, including vitamins (B1, B2 and B3), ash content, carbohydrates, fat content, moisture content, protein content, as well as levels of magnesium, phosphorus, potassium, sodium and zinc.

### Analysis of chemical profiles of Kaab Dum rice extract

Chemical component separation was performed using an Agilent 1260 Infinity Series HPLC-DAD System (Agilent Technologies, Waldbronn, Germany) equipped with a Luna C-18 (2) column (4.6 × 150 mm, 5 μm, Phenomenex Inc., Torrance, CA, USA). The column temperature was maintained at 35˚C. The mobile phase consisted of water (A) and acetonitrile (B), both containing 0.1% (v/v) formic acid. The gradient elution profile started with 5% acetonitrile (B) for 30 sec, followed by a gradual increase in acetonitrile concentration to 95% over a period of 30 sec and then held at 95% acetonitrile for 10 sec. The flow rate was set at 0.5 mL/min, and the injection volume was 10 μL. The molecular weight analysis of substances was carried out using a 6540 Ultra-High-Definition Accurate Q-TOF-mass spectrometer (Agilent Technologies, Singapore) with ionization through electrospray ionization (ESI). The molecular weight analysis range was m/z 100–1000 Da. The instrument conditions for molecular weight analysis included a nitrogen gas (N2) flow rate of 7 L/min, a temperature of 350˚C, a nebulizer gas pressure of 30 psi, a capillary voltage of 3.5 kV, fragmentation at 100 V, skimmer voltage at 65 V, Vcap at 3500 V, octopole RFP at 750 V, and collision energy at 10, 20, and 40 V using ultra-pure nitrogen gas (99.9995%). The compound types were analyzed using Agilent MassHunter Data Acquisition Software, Version B.05.01, and Agilent MassHunter Qualitative Analysis Software B.06.00 (Agilent Technologies, Santa Clara, CA, USA), in combination with databases such as METLIN PCD/PCDL database (Agilent Technologies, Santa Clara, CA, USA) and the public database Human Metabolome Database (http://www.hmdb.ca, accessed on 19 December 2022).

### Cell culture experiments

**Preparation of the extract.** The Kaab Dum rice extract was dissolved in dimethyl sulfoxide (DMSO) to 0.1 g/ml as a stock concentration and mixed thoroughly using a vortex mixer. Subsequently, it was filtered through a 0.45 μm filter, stored at -20˚C and protected from light.

**HaCaT keratinocyte culture.** HaCaT cells obtained from the American Type Culture Collection (ATCC) were cultured in Dulbecco's Modified Eagle's Medium (DMEM) with high glucose concentration, supplemented with 10% fetal bovine serum and 1% w/v penicillin/ streptomycin. The cells were then incubated in a controlled humidity incubator at 37°C under a 5% $CO_2$ atmosphere.

**Assessment of cell viability using 3-(4,5- dimethylthiazol-2-yl)-2,5-diphenyl-tetrazo-lium bromide (MTT) assay.** HaCaT cells were cultured in 96-well plates for 24 h. Then, they were exposed to different concentrations of the polished and unpolished Kaab Dum rice extract. The cells were then further incubated for 24 h at 37°C in a 5% $CO_2$ atmosphere. Following incubation, a media containing MTT solution with a concentration of 0.5 mg/ml was added to each well, with a volume of 100 μL per well. The cells were incubated for an additional 2 h to allow the formation of purple formazan crystals in viable cells. Subsequently, the cells were dissolved using DMSO at a volume of 100 μL per well to measure the absorbance of the formazan solution, which directly correlates with the number of viable cells. The absorbance was measured at a wavelength of 570 nm using an automatic microplate reader (EMax® Plus Microplate Reader, Molecular devices) [17].

**Detection of cellular oxidative stress using the 2,7-dichlorodihydrofluorescein diacetate (DCFH-DA) staining.** To assess cellular oxidative stress status, the DCFH-DA method was employed. HaCaT cells were cultured in 96-well plates at a density of $1.5 \times 10^4$ cells per well in completed media with a volume of 200 μL. The cells were then incubated in a $CO_2$ incubator at 37°C for 24 h to allow them to adhere to the well plate. Subsequently, the culture medium was removed, and chronic wound conditions were simulated by adding transforming growth factor-β (TGF-β: Abbkine, PRP100190) at a concentration of 10 ng/ml for 48 h. Then, the cells were treated with polished Kaab Dum rice extract at concentrations of 0.02 and 0.1 mg/ml, as well as with unpolished Kaab Dum rice extract at concentrations of 0.2 and 1 mg/ml, and further incubated for 24 h in a volume of 200 μL. Afterward, the culture medium was removed and replaced with fresh media containing DCFH-DA at a concentration of 10 μmol, and the cells were incubated for another hour. Following incubation, the level of fluorescence representing oxidative stress was measured at an excitation/emission wavelength of 485/535 nm using a Fluorescence microplate reader (The Spark® multimode microplate reader, TECAN, Switzerland) [18].

**Detection of ER stress using fluorescent technique.** To investigate the expression of PERK protein in HaCaT cells, the cells were cultured in 6-well plates containing coverslips for 24 h. The cells were then incubated in a $CO_2$ incubator at 37°C for an additional 24 h to allow them to adhere to the cover slips. Subsequently, chronic wound conditions were simulated by adding TGF-β at a concentration of 10 ng/ml for 48 h. Then, the cells were treated with polished Kaab Dum rice extract at concentrations of 0.02 and 0.1 mg/ml, as well as with unpolished Kaab Dum rice extract at concentrations of 0.2 and 1 mg/ml, and further incubated for 24 h. Following incubation, the cells were fixed and then subjected to immunostaining. The primary antibody targeting PERK protein (Abcam: ab65142) was applied, followed by a secondary antibody. The cell nuclei were stained with DAPI to visualize protein expression under a fluorescence microscope (Nikon ECLIPSE Ni, Japan). Five fields were randomly selected and photographed with a digital camera (Nikon Digital Sight 10, Japan) using NIS-Elements software and measured for the mean intensity of PERK using ImageJ software (National Institutes of Health, Bethesda, Maryland, USA).

**Wound healing assay.** HaCaT cells were cultured for 24 h in 6-well plates until they reached approximately 70% confluence. Subsequently, a pipette with a size of 200 μL was used to create a wound between the cells to simulate the formation of chronic wounds. This was followed by the addition of TGF-β at a concentration of 10 ng/ml for 48 h to induce chronic

wound conditions. Then, the cells were treated with polished Kaab Dum rice extract at concentrations of 0.02 and 0.1 mg/ml, as well as with unpolished Kaab Dum rice extract at concentrations of 0.2 and 1 mg/ml. The cells were then incubated for an additional 24 and 48 h. Subsequently, images were captured using an inverted phase contrast microscope (CKX3, Olympus, Japan) at 24 and 48 h, and the extent of cell migration was compared between the group treated with Kaab Dum rice extract and the chronic wound control group [19].

**Detection of cell-cycle arrest using flow cytometry.** Cultured HaCaT cells in 6-well plates and incubated them in a $CO_2$ incubator at 37˚C with 5% $CO_2$ for 24 h. Simulated chronic wound conditions by adding TGF-β at a concentration of 10 ng/ml for 48 h. Then, the cells were treated with polished Kaab Dum rice extract at concentrations of 0.02 and 0.1 mg/ml, as well as with unpolished Kaab Dum rice extract at concentrations of 0.2 and 1 mg/ml. Subsequently, transferred the cells back to the same incubator conditions for 24 h. Then, washed the cells with Phosphate buffer saline (PBS) and detached them from the 6-well plates using trypsinization. Collected the cells in microtubes and centrifuged them at 1,200 rpm for 5 min. Then, re-suspended the cells in binding buffer with a volume of 200 μL and stained them with anti-p21 antibody (Invitrogen: 33–7000) at a volume of 2 μL for 15 min and stained with secondary antibody conjugated with Alexa Fluor® 488 (Abcam, ab150077). Incubated the mixture at room temperature in the dark for 10 min. Finally, quantified the expression levels of p21 protein using FACScan (Becton, Dickinson and Company, Franklin Lakes, NJ, USA).

**Detection of cellular senescence using senescence-associated β-galactosidase (SA-β-gal) staining.** HaCaT cells were cultured for 24 h in 6-well plates. Chronic wound conditions were simulated by adding TGF-β at a concentration of 10 ng/ml for 48 h. Subsequently, the cells were treated with polished Kaab Dum rice extract at concentrations of 0.02 and 0.1 mg/ml, as well as with unpolished Kaab Dum rice extract at concentrations of 0.2 and 1 mg/ml, and incubated for an additional 24 h. Afterward, the cells were fixed using 0.2% glutaraldehyde as a fixative reagent and washed with PBS. They were then stained with SA-β-gal enzyme using SA-β-gal staining (Sigma, CAS 7240-90-6) for 16 h at 37˚C, in the absence of $CO_2$ and light. The cells were examined under a microscope, and a comparison was made between the group treated with rice extract and the chronic wound control group.

## Statistical analysis

All data are presented as mean ± standard deviation (Mean ± SD). Differences between the means of all test groups were analyzed using one-way analysis of variance (ANOVA) and pairwise comparisons of means were performed using Tukey's multiple comparison test with a confidence level of 95% (p-value < 0.05) using Prism version 9 software.

## Results

### The nutritional compositions of Kaab Dum rice extracts

The analysis encompassed various nutritional components, including vitamin B1, B2, B3, ash, carbohydrates, fat, moisture, protein, magnesium, phosphorus, potassium, sodium and zinc. The results revealed that unpolished Kaab Dum rice extract exhibits higher levels of vitamin B1, B2, B3, fat, protein, iron, magnesium, phosphorus and potassium compared to polished rice extract, as detailed in Table 1.

### Chemical profiles of Kaab Dum rice extract

Chemical composition analysis of polished Kaab Dum rice extract revealed the presence of various compounds, including sugars (3-beta-Cellobiosylglucose and D-fructose, 6-O-alpha-

**Table 1. The nutritional compositions of extracts from polished and unpolished Kaab Dum rice.**

| Compositions | Unit | Polished Kaab Dum rice | Unpolished Kaab Dum rice |
|---|---|---|---|
| Vitamin B1 (Thiamine) | mg/100g | 0.078 | 0.124 |
| Vitamin B2 (Riboflavin) | mg/100g | 0.011 | 0.019 |
| Vitamin B3 (Nicotinic acid) | mg/100g | 0.265 | 0.429 |
| Ash | g/100g | 1.1 | 1.56 |
| Carbohydrate | g/100g | 76.6 | 74.9 |
| Fat | g/100g | 1.28 | 2.03 |
| Moisture | g/100g | 12.69 | 12.77 |
| Protein | g/100g | 8.33 | 8.74 |
| Copper | mg/kg | 0.397 | 0.266 |
| Iron | mg/kg | 5.349 | 8.046 |
| Magesium | mg/kg | 579.73 | 1,070.26 |
| Phosphorus | mg/kg | 1,581.99 | 2,623.27 |
| Potassium | mg/kg | 1,273.87 | 2,162.94 |
| Sodium | mg/kg | 32.51 | 28.67 |
| Zinc | mg/kg | 27.395 | 24.553 |

D-glucopyranosyl-), lipids (glycerophosphocholine, C16 sphinganine, dehydrophytosphingo-sine, 1-myristoyl-glycero-3-phosphocholine, 1-(9Z,12Z,15Z-octadecatrienoyl)-sn-glycero-3-phosphocholine, (9Z,12Z-octadecadienoyl)-lysophosphatidylethanolamine, 2-(9Z,12Z-octa-decadienoyl)-sn-glycero-3-phosphocholine, 1-eicosapentaenoyl-glycero-3-phosphocholine, glycerophospho-N-palmitoyl ethanolamine, 1-palmitoyl phosphatidylcholine, and linoleoyl ethanolamide), an amino alcohol (2-amino-3-methyl-1-butanol), an essential amino acid (tryptophan) and choline (palmitoylcholine), as depicted in Fig 1A and detailed in Table 2.

Similarly, the chemical composition analysis of unpolished Kaab Dum rice extract detected the presence of lipids (dehydrophytosphingosine, 4E,14Z-Sphingadiene, 1-myristoyl-glycero-3-phosphocholine, 1-(9Z,12Z,15Z-octadecatrienoyl)-sn-glycero-3-phosphocholine, (9Z,12Z-octadecadienoyl)-lysophosphatidylethanolamine, 2-(9Z,12Z-octadecadienoyl)-sn-glycero-3-phosphocholine, 1-eicosapentaenoyl-glycero-3-phosphocholine, glycerophospho-N-palmi-toyl ethanolamine, and 1-palmitoyl phosphatidylcholine), an amino alcohol (2-amino-3-methyl-1-butanol), an essential amino acid (tryptophan), a benzylamino group (10-benzyla-mino-1-decanol) and (S)-Nerolidol 3-O-[a-L-rhamnopyranosyl-(1->4)-a-L-rhamnopyrano-syl-(1->2)-b-D-glucopyranoside]. These components are depicted in Fig 1B and detailed in Table 3.

### Cytotoxicity of Kaab Dum rice extract on HaCaT cells

The results demonstrated that polished Kaab Dum rice extract, tested at concentrations of 0.001, 0.01, 0.1, 1 and 10 mg on HaCaT cells, resulted in % cell viabilities of 117.59 ± 13.80, 92.85 ± 15.12, 81.00 ± 6.31, 67.12 ± 5.68 and 3.99 ± 0.77, respectively. Notably, a concentration of 0.1 mg was identified as a safe range for investigating effects of polished Kaab Dum rice extract on the cells (Fig 2A). Toxicity assessments of polished Kaab Dum rice extract at concentrations ranging from 0.02 to 0.1 mg revealed % cell viabilities of 95.26 ± 18.66, 97.44 ± 7.46, 97.89 ± 2.69, 92.34 ± 6.40 and 88.11 ± 8.58, respectively. These findings indicate that the highest safe concentration of polished Kaab Dum rice extract did not induce toxicity in HaCaT cells, as illustrated in the Fig 2B.

Similarly, unpolished Kaab Dum rice extract, tested at concentrations of 0.001, 0.01, 0.1, 1 and 10 mg on HaCaT skin cells, resulted in % cell viabilities of 107.78 ± 7.74, 112.37 ± 11.14,

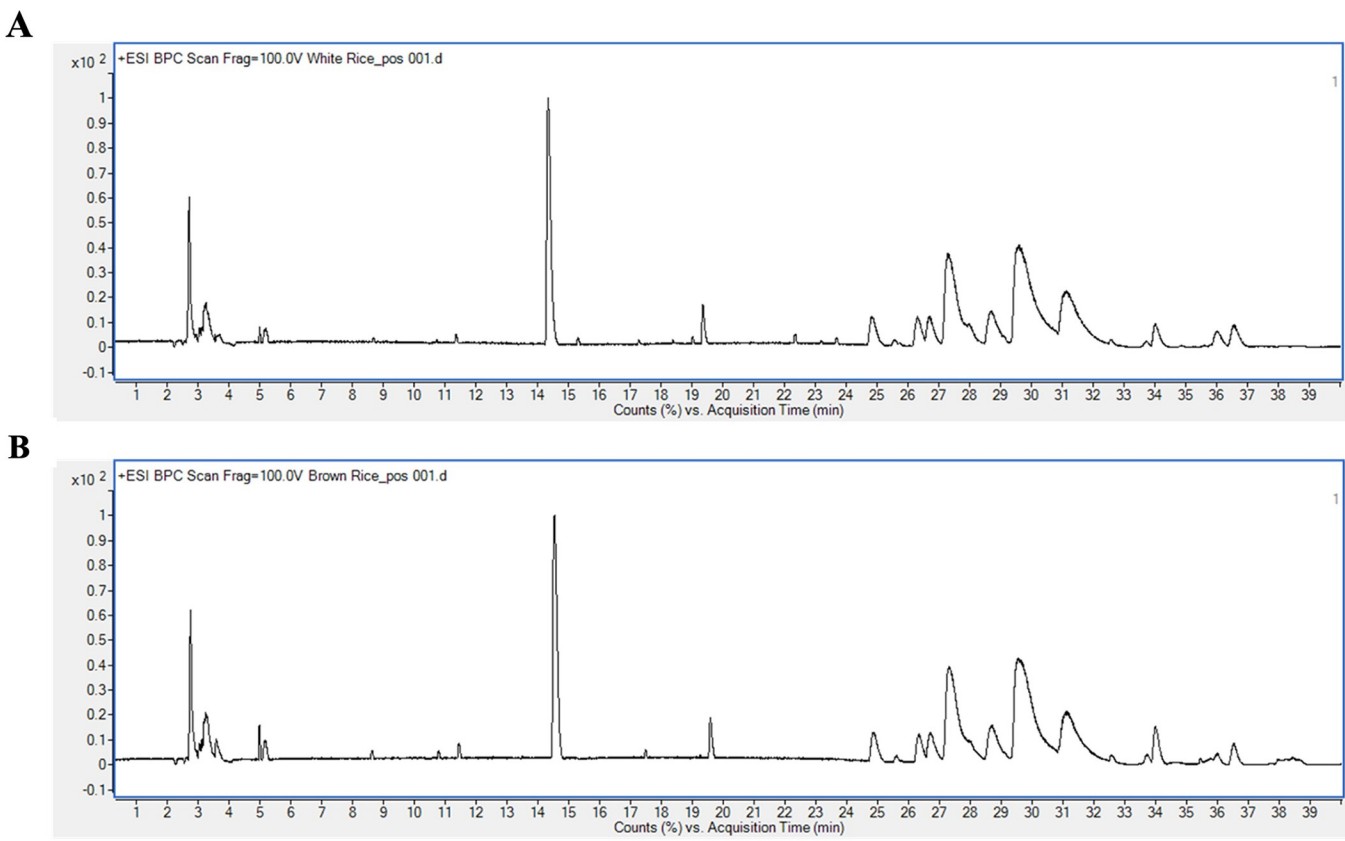

**Fig 1.** LC-ESI-QTOF-MS/MS chromatograms of polished (A) and unpolished (B) Kaab Dum rice extracts.

111.05 ± 2.17, 80.76 ± 8.97 and 4.43 ± 0.48, respectively. A concentration of 1 mg was identified as a safe range for studying the effects of unpolished Kaab Dum rice extract on the cells (Fig 2C). Toxicity assessments of brown Kaab Dum rice extract at concentrations ranging from 0.2 to 1 mg showed % cell viabilities of 100.86 ± 13.45, 100.26 ± 26.44, 89.68 ± 15.06, 88.09 ± 22.42 and 86.12 ± 15.77, respectively. These results indicate that the highest safe concentration of unpolished Kaab Dum rice extract did not induce toxicity in HaCaT skin cells, as depicted in the Fig 2D.

### Effect of Kaab Dum rice extract on oxidative stress in HaCaT cells

The assessment of oxidative stress induced by polished Kaab Dum rice extract was conducted using the 2,7-dichlorodihydrofluorescein diacetate (DCFH-DA) method. The test groups were categorized as follows: the control group, the TGF-β wound group (10 ng/ml TGF-β), the wound group treated with low concentration of polished Kaab Dum rice extract (10 ng/ml TGF-β + 0.02 mg/ml PKD), and the wound group treated with high concentration of polished Kaab Dum rice extract (10 ng/ml TGF-β + 0.1 mg/ml PKD). After 24 h, the assessment of cellular oxidant levels in HaCaT cells showed values of 128.21 ± 4.44, 90.78 ± 1.44 and 76.78 ± 10.48, respectively, compared to the control (100%). In all test groups, cellular oxidant levels in the skin cells were significantly reduced, with statistical significance at $p < 0.01$, $p < 0.001$ and $p < 0.0001$, respectively, when compared to the wound group. For unpolished Kaab Dum rice extract, the test groups were categorized as follows: the control group, the TGF-β wound group (10 ng/ml TGF-β), the wound group treated with low concentration of unpolished Kaab Dum

**Table 2. Chemical compositions of polished Kaab Dum rice extract.**

| RT (min) | m/z | Adduct | MS/MS | Tentative Identification | Formula | Error (ppm) |
|---|---|---|---|---|---|---|
| 2.753 | 104.1072 | [M+H]+ | 60.0807,58.0651 | 2-Amino-3-methyl-1-butanol | $C_5H_{13}NO$ | -2.01 |
| 3.057 | 258.1105 | [M+H]+ | 184.0733,104.1067,60.0807 | Glycerophospho choline | $C_8H_{20}NO_6P$ | -1.55 |
| 3.115 | 527.1592 | [M+Na]+ | 365.1050,277.0834,203.0522,104.1069 | 3-beta-Cellobiosylglucose | $C_{18}H_{32}O_{16}$ | -1.79 |
| 3.258 | 365.1062 | [M+Na]+ | 289.0492,271.0408,247.0402,229.0287,203.0523,185.0419 | D-Fructose, 6-O-alpha-D-glucopyranosyl- | $C_{12}H_{22}O_{11}$ | -2.1 |
| 8.629 | 205.0975 | [M+H]+ | 188.0702,146.0596 | (±)-Tryptophan | $C_{11}H_{12}N_2O_2$ | -1.69 |
| 14.541 | 325.2283 | [M+H]+ | 233.1649,148.1119,86.0965 | Diampromide | $C_{21}H_{28}N_2O$ | -2.64 |
| 17.486 | 274.2748 | [M+H]+ | 198.9290,88.0756 | C16 Sphinganine | $C_{16}H_{35}NO_2$ | -2.71 |
| 18.608 | 316.2852 | [M+H]+ | 143.0010,60.0444 | Dehydrophyto sphingosine | $C_{18}H_{37}NO_3$ | -1.83 |
| 23.409 | 342.3374 | [M+H]+ | 283.2633,57.0701 | Palmitoylcholine | $C_{21}H_{44}NO_2$ | -0.57 |
| 24.205 | 468.3084 | [M+H]+ | 184.0732,85.0968 | 1-Myristoyl-glycero-3-phosphocholine | $C_{22}H_{46}NO_7P$ | 0.14 |
| 25.118 | 518.3246 | [M+H]+ | 459.2494,184.0731,104.1067 | 1-(9Z,12Z,15Z-Octadecatrienoyl)-sn-glycero-3-phosphocholine | $C_{26}H_{48}NO_7P$ | -0.93 |
| 26.343 | 478.2939 | [M+H]+ | 337.2736,95.0849 | (9Z,12Z-Octadecadienoyl)-lysophosphatidylethanolamine | $C_{23}H_{44}NO_7P$ | -2.27 |
| 26.714 | 520.3413 | [M+H]+ | 461.2046,337.2734,258.1097,184.0733,86.0963 | 2-(9Z,12Z-Octadecadienoyl)-sn-glycero-3-phosphocholine | $C_{26}H_{50}NO_7P$ | -2.95 |
| 27.37 | 542.3231 | [M+H]+ | 483.2480,337.2740,199.1695,104.1067 | 1-Eicosapentaenoyl-glycero-3-phosphocholine | $C_{28}H_{48}NO_7P$ | 1.87 |
| 27.991 | 454.2939 | [M+H]+ | 313.2736,216.0634,155.0103,62.0599 | Glycerophospho-N-palmitoyl ethanolamine | $C_{21}H_{44}NO_7P$ | -2.39 |
| 28.696 | 496.3415 | [M+H]+ | 313.2745,258.1102,184.0734,86.0964 | 1-Palmitoylphosphati dylcholine | $C_{24}H_{50}NO_7P$ | -3.49 |
| 29.035 | 480.3094 | [M+H]+ | 339.2891,216.0638,155.0105,62.0603 | Glycerophospho-N-oleoyl ethanolamine | $C_{23}H_{46}NO_7P$ | -1.95 |
| 32.228 | 324.2901 | [M+H]+ | 306.2796,62.0599 | Linoleoyl ethanolamide | $C_{20}H_{37}NO_2$ | -1.22 |

rice extract (10 ng/ml TGF-β + 0.2 mg/ml UKD), and the wound group treated with high concentration of unpolished Kaab Dum rice extract (10 ng/ml TGF-β + 1 mg/ml UKD). After 24 h, the assessment of cellular oxidative stress in HaCaT cells showed values of 128.21 ± 4.44, 90.90 ± 4.01 and 85.61 ± 6.03, respectively, compared to the control (100%). In all test groups, cellular oxidative stress in the skin cells was significantly reduced, with statistical significance at $p < 0.001$ and $p < 0.0001$, respectively, when compared to the wound group (Fig 3).

### Inhibition of ER stress induction by Kaab Dum rice extract

The results of the inhibition of ER stress induction by Kaab Dum rice extract, as assessed by monitoring the expression of PERK protein in HaCaT cells using the immunofluorescence technique. For polished Kaab Dum rice extract, the test groups were categorized as the control group, the TGF-β wound group (10 ng/ml TGF-β), the wound group treated with low concentration polished Kaab Dum rice extract (10 ng/ml TGF-β + 0.02 mg/ml PKD), and the wound group treated with high concentration polished Kaab Dum rice extract (10 ng/ml TGF-β + 0.1 mg/ml PKD). After 24 h, the assessment of PERK protein expression in HaCaT cells revealed ER stress values of 3.59 ± 0.02, 12.51 ± 1.17, 7.32 ± 0.34 and 3.65 ± 0.21, respectively. In all test groups, the induction of ER stress in skin cells was significantly reduced, with statistical significance at $p < 0.0001$ when compared to the wound group. For unpolished Kaab Dum rice extract, the test groups were categorized as the control group, the TGF-β wound group (10 ng/ml TGF-β), the wound group treated with low concentration unpolished Kaab Dum rice extract (10 ng/ml TGF-β + 0.2 mg/ml UKD), and the wound group treated with high concentration unpolished Kaab Dum rice extract (10 ng/ml TGF-β + 1 mg/ml UKD). After 24 h, the

**Table 3. Chemical compositions of unpolished Kaab Dum rice extract.**

| RT (min) | m/z | Adduct | MS/MS | Tentative Identification | Formula | Error (ppm) |
|---|---|---|---|---|---|---|
| 2.713 | 104.1072 | [M+H]+ | 60.0807,58.0651 | 2-Amino-3-methyl-1-butanol | $C_5H_{13}NO$ | -2.01 |
| 8.687 | 205.0973 | [M+H]+ | 188.0704,146.0597,74.0241 | (±)-Tryptophan | $C_{11}H_{12}N_2O_2$ | -0.71 |
| 14.349 | 325.2285 | [M+H]+ | 233.1648,148.1116,91.0544,86.0965 | Diampromide | $C_{21}H_{28}N_2O$ | -3.26 |
| 15.313 | 264.2327 | [M+H]+ | 219.1741,203.1429 | 10- Benzylamino-1-decanol | $C_{17}H_{29}NO$ | -1.55 |
| 18.391 | 316.2853 | [M+H]+ | 194.9154,60.0444 | Dehydrophytosphingosine | $C_{18}H_{37}NO_3$ | -2.15 |
| 19.028 | 298.2748 | [M+H]+ | 280.2633,67.0543 | 4E,14Z-Sphingadiene | $C_{18}H_{35}NO_2$ | -2.49 |
| 24.847 | 490.2915 | [M+Na]+ | 431.2163,285.2424,104.1067 | 1-Myristoyl-glycero-3-phosphocholine | $C_{22}H_{46}NO_7P$ | -2.22 |
| 25.096 | 518.3249 | [M+H]+ | 459.2471,335.2589,258.1095,184.0730,104.1066 | 1-(9Z,12Z,15Z-Octadecatrienoyl)-sn-glycero-3-phosphocholine | $C_{26}H_{48}NO_7P$ | -1.51 |
| 26.325 | 478.2942 | [M+H]+ | 417.2388,337.2735,263.2365,155.0101,62.0599 | (9Z,12Z-Octadecadienoyl)-lysophosphatidylethanolamine | $C_{23}H_{44}NO_7P$ | -2.89 |
| 26.704 | 520.3414 | [M+H]+ | 337.2731,184.0734,86.0961 | 2-(9Z,12Z-Octadecadienoyl)-sn-glycero-3-phosphocholine | $C_{26}H_{50}NO_7P$ | -3.14 |
| 26.704 | 542.3227 | [M+H]+ | 483.2478,337.2734,146.9815,86.0962 | 1-Eicosapentaenoyl-glycero-3-phosphocholine | $C_{28}H_{48}NO_7P$ | 2.61 |
| 27 | 677.3735 | [M+H]+ | 515.3195,347.0896 | (S)-Nerolidol 3-O-[a-L-Rhamnopyranosyl-(1->4)-a-L-rhamnopyranosyl-(1->2)-b-D-glucopyranoside] | $C_{33}H_{56}O_{14}$ | 1.16 |
| 27.361 | 542.3231 | [M+H]+ | 483.2480,337.2739,104.1067 | 1-Eicosapentaenoyl-glycero-3-phosphocholine | $C_{28}H_{48}NO_7P$ | 1.87 |
| 27.992 | 454.2942 | [M+H]+ | 393.2399,313.2737,155.0104,62.0599 | Glycerophospho-N-palmitoyl ethanolamine | $C_{21}H_{44}NO_7P$ | -3.05 |
| 28.708 | 496.3415 | [M+H]+ | 313.2733,258.1099,184.0735,86.0962 | 1-Palmitoylphosphatidylcholine | $C_{24}H_{50}NO_7P$ | -3.49 |
| 29.04 | 480.3094 | [M+H]+ | 339.2887,155.0082,62.0597 | Glycerophospho-N-oleoyl ethanolamine | $C_{23}H_{46}NO_7P$ | -1.95 |

assessment of PERK protein expression in HaCaT cells revealed ER stress values of 3.59 ± 0.02, 12.51 ± 1.17, 9.22 ± 0.20 and 6.67±0.78, respectively. In all test groups, the induction of ER stress in skin cells was significantly reduced, with statistical significance at $p < 0.01$ and $p < 0.0001$ when compared to the wound group (Fig 4A and 4B).

## Effect of Kaab Dum rice extract on the migration (wound closure) of HaCaT cells induced as chronic wounds by TGF-β

For polished Kaab Dum rice extract, the test groups were divided into the following groups: the control group, the TGF-β-induced wound group (10 ng/ml TGF-β), the low-concentration polished Kaab Dum rice extract-treated wound group (10 ng/ml TGF-β + 0.02 mg/ml PKD), and the high-concentration polished Kaab Dum rice extract-treated wound group (10 ng/ml TGF-β + 0.1 mg/ml PKD). After 24 h, cell migration (wound closure) was measured, and the percentages of cell migration for the test groups were found to be 25.89 ± 7.73, 76.30 ± 4.28, and 102.16 ± 15.31, respectively, compared to the control group (100%). All test groups exhibited a statistically significant increase in cell migration compared to the wound group (Fig 5A and 5B). Furthermore, after 48 h, cell migration (wound closure) was measured again, and the percentages

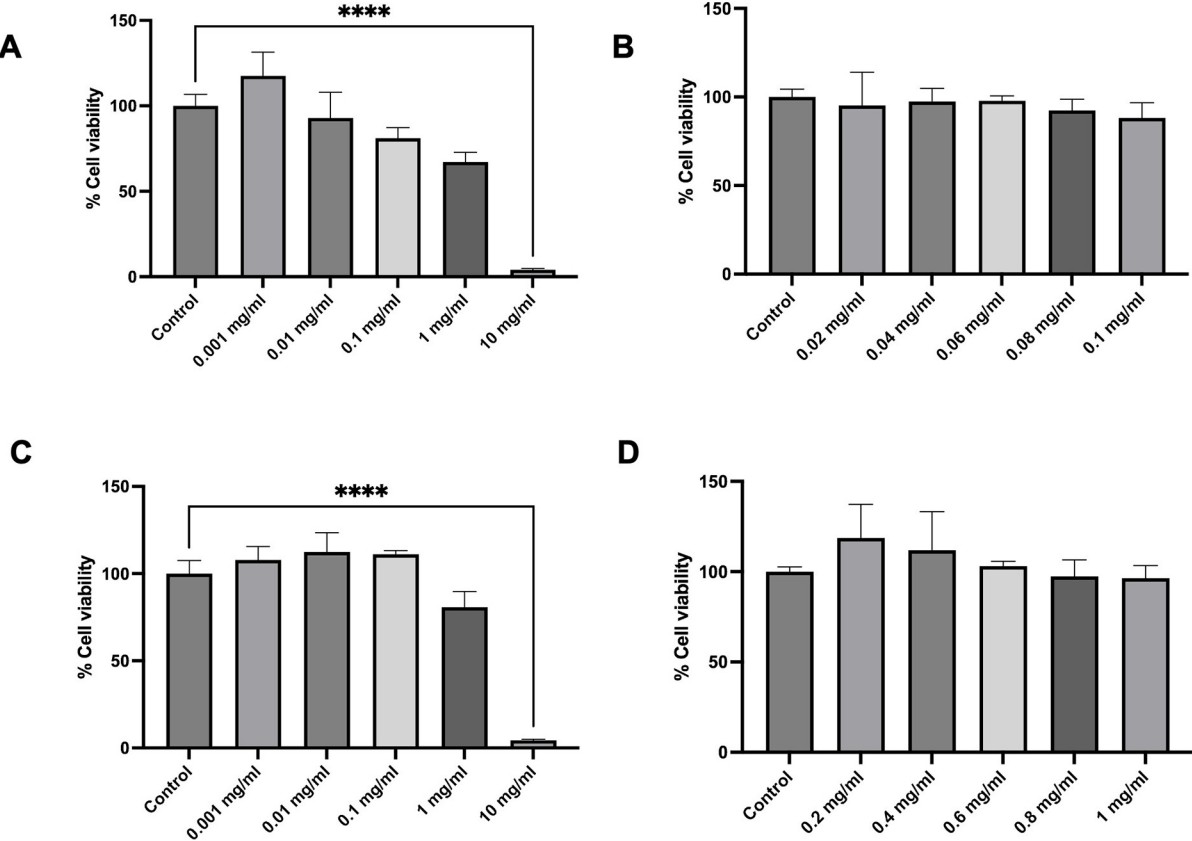

**Fig 2. Cytotoxicity of Kaab Dum rice extracts on HaCaT cells using the MTT assay for 24 h.** A and C: Safety dose for polished and unpolished extract, respectively. B and D: The highest safe concentration dose for polished and unpolished extract, respectively. Significance level: **** ($p < 0.0001$).

of cell migration for the test groups were found to be 51.14 ± 8.15, 87.79 ± 2.63 and 119.64 ± 8.62, respectively, compared to the control group. Once again, all test groups showed a statistically significant increase in cell migration compared to the wound group (Fig 5A and 5C).

For unpolished Kaab Dum rice extract, the test groups were divided into the following groups: the control group, the TGF-β-induced wound group (10 ng/ml TGF-β), the low-concentration unpolished Kaab Dum rice extract-treated wound group (10 ng/ml TGF-β + 0.2 mg/ml UKD), and the high-concentration unpolished Kaab Dum rice extract-treated wound group (10 ng/ml TGF-β + 1 mg/ml UKD). After 24 h, the percentages of cell migration for the test groups were found to be 41.54 ± 8.70, 56.79 ± 15.15, and 68.48 ± 8.90, respectively, compared to the control group (100%). All test groups exhibited a statistically significant increase in cell migration compared to the wound group (Fig 5A and 5B). Furthermore, after 48 h, cell migration (wound closure) was measured again, and the percentages of cell migration for the test groups were found to be 46.29 ± 1.46, 52.36 ± 4.34 and 70.58 ± 8.89, respectively, compared to the control group. Once again, all test groups showed a statistically significant increase in cell migration compared to the wound group (Fig 5A and 5C).

## Effect of Kaab Dum rice extract on the expression of p21 protein

For polished Kaab Dum rice extract, the inhibition of p21 protein expression, which is related to cell cycle arrest was assessed using the flow cytometry. The test groups were categorized as

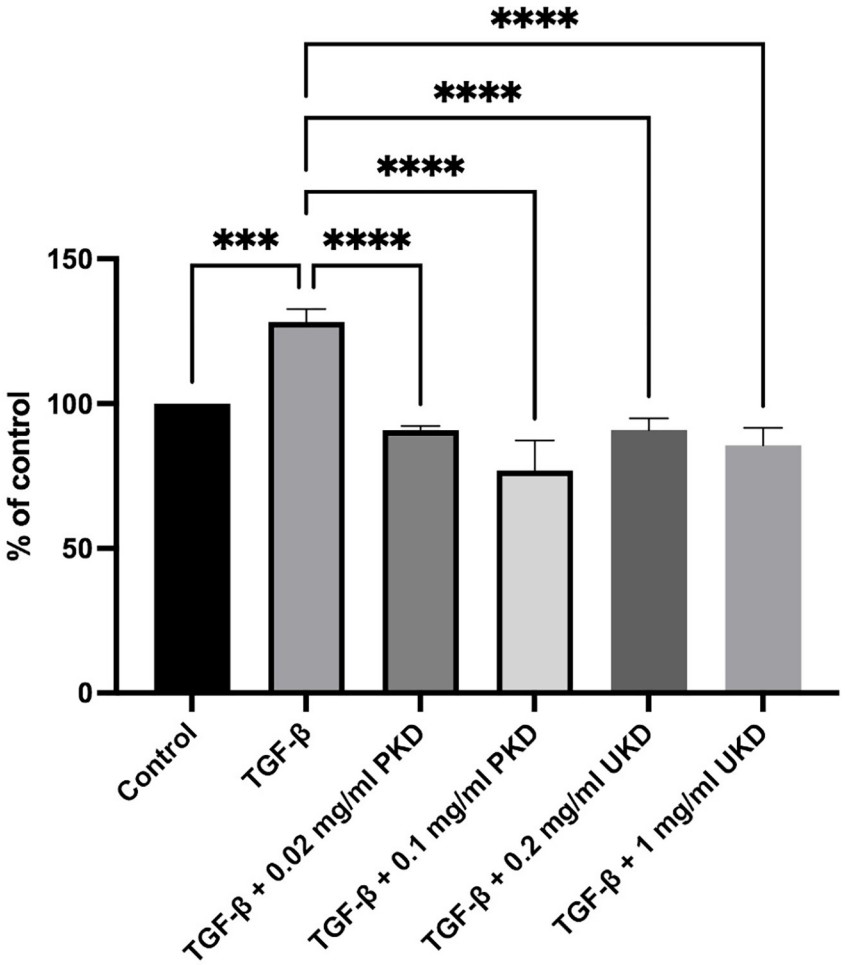

**Fig 3. Effect of polished (PKD) and unpolished (UKD) Kaab Dum rice extracts on oxidative stress in TGF-β-induced HaCaT cells using the DCFH-DA method.** Oxidative stress represents 100% control. Statistical significance: *** ($p < 0.001$), **** ($p < 0.0001$).

follows: the control group, the TGF-β wound group (10 ng/ml TGF-β), the wound group treated with low-concentration polished rice extract (10 ng/ml TGF-β + 0.02 mg/ml PKD), and the wound group treated with high-concentration polished rice extract (10 ng/ml TGF-β + 0.1 mg/ml PKD). After 24 h, the measurement of p21 protein expression in HaCaT cells revealed the percentage of positive cells as follows: 3.99 ± 0.46, 34.29 ± 1.10, 10.24 ± 1.11 and 3.45 ± 0.53, respectively. All test groups exhibited a statistically significant reduction in p21 protein expression in skin cells compared to the wound group.For unpolished Kaab Dum rice extract, the test groups were divided into: the control group, the TGF-β wound group (10 ng/ml TGF-β), the wound group treated with low-concentration unpolished rice extract (10 ng/ml TGF-β + 0.2 mg/ml UKD), and the wound group treated with high-concentration unpolished rice extract (10 ng/ml TGF-β + 1 mg/ml UKD). After 24 h, the measurement of p21 protein expression in HaCaT cells revealed the percentage of positive cells as follows: 3.99 ± 0.46, 34.29 ± 1.10, 11.71 ± 1.87 and 8.20 ± 1.51, respectively. All test groups showed a statistically significant reduction in p21 protein expression in skin cells compared to the wound group (Fig 6A and 6B).

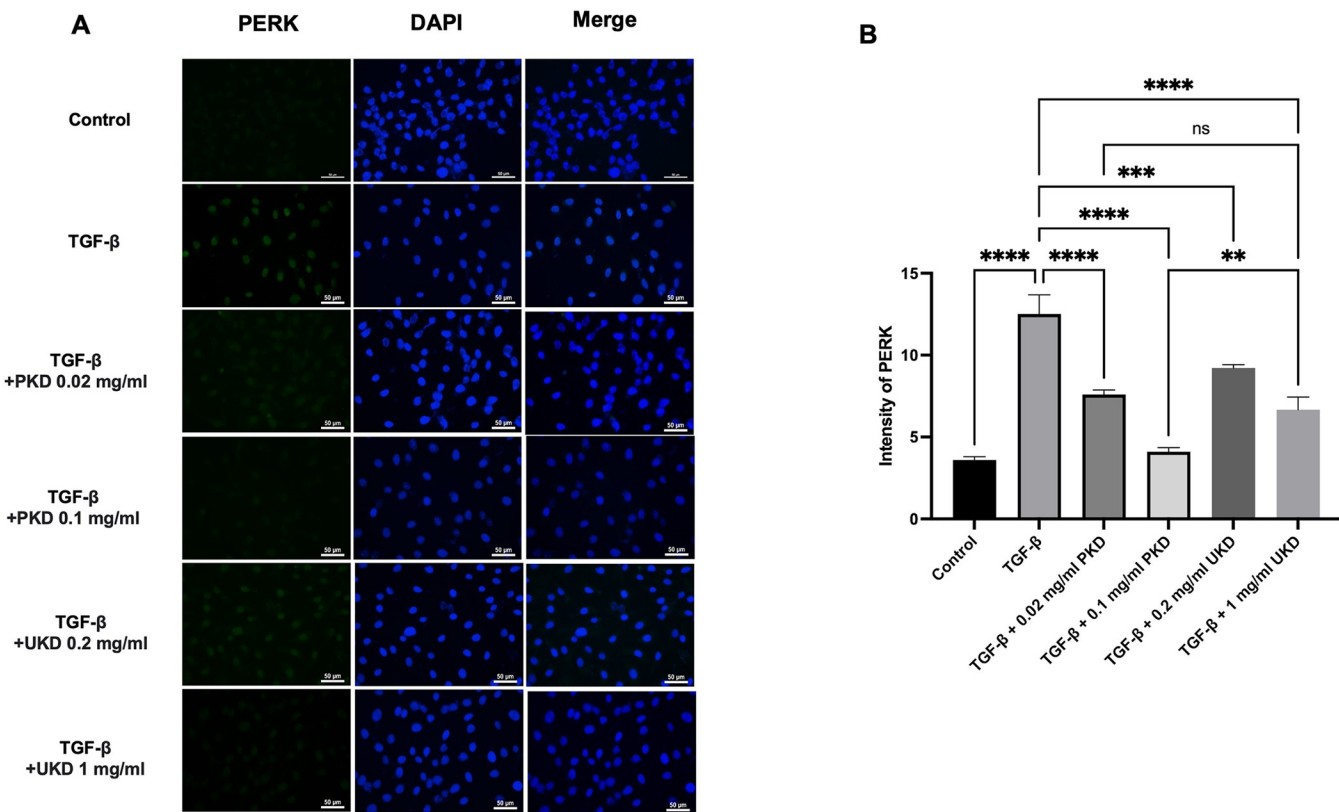

**Fig 4. Effect of polished (PKD) and unpolished (UKD) Kaab Dum rice extracts on ER stress inhibition in HaCaT cells induced as chronic wounds by TGF-β, observed under a fluorescence microscope at 40X magnification.** A: Expression of PERK protein after 24 h. B: Intensity of PERK protein after 24 h. Statistical significance: ** (p < 0.01), **** (p < 0.0001).

## Effects of Kaab Dum rice extract on senescence of HaCaT cells induced as chronic wounds by TGF-β

For polished Kaab Dum rice extract, the test groups were categorized as follows: the control group, the TGF-β-induced wound group (10 ng/ml TGF-β), the low-concentration polished Kaab Dum rice extract-treated wound group (10 ng/ml TGF-β + 0.02 mg/ml PKD), and the high-concentration polished Kaab Dum rice extract-treated wound group (10 ng/ml TGF-β + 0.1 mg/ml PKD). After 24 h, the percentage of positively stained blue cells, indicative of senescence due to the SA-β-gal enzyme reaction in HaCaT skin cells, was determined to be 4.67 ± 1.15, 64.33 ± 8.50, 30.67 ± 5.69 and 14.00 ± 6.24, respectively. All test groups exhibited a statistically significant reduction in the presence of blue-stained senescent cells compared to the wound group. For unpolished Kaab Dum rice extract, following a similar approach, the test groups were divided into: the control group, the TGF-β-induced wound group (10 ng/ml TGF-β), the low-concentration unpolished Kaab Dum rice extract-treated wound group (10 ng/ml TGF-β + 0.2 mg/ml UKD), and the high-concentration unpolished Kaab Dum rice extract-treated wound group (10 ng/ml TGF-β + 1 mg/ml UKD). After 24 hours, the percentage of positively stained blue cells, reflecting senescence due to the SA-β-gal enzyme reaction in HaCaT skin cells, was measured and found to be 6.33 ± 3.06, 62.00 ± 18.33, 34.67 ± 11.02 and 20.33 ± 4.73, respectively. All test groups exhibited a statistically significant reduction in blue-stained senescent cells compared to the wound group (Fig 7A and 7B).

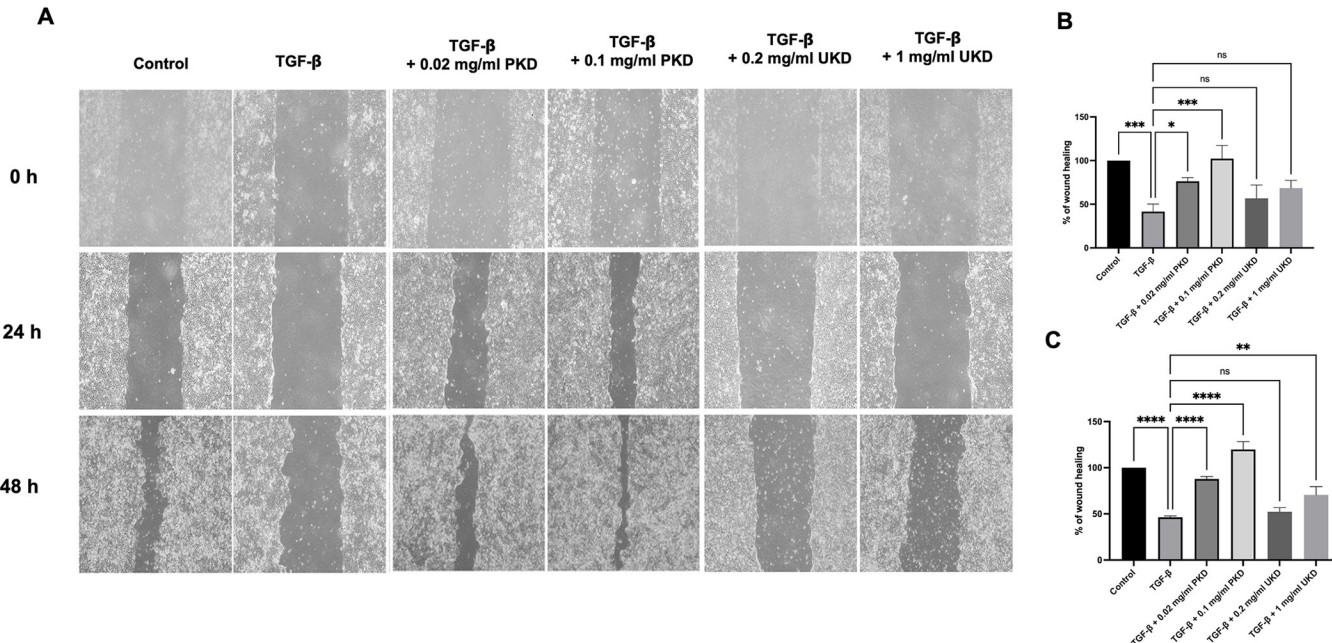

**Fig 5. Effect of polished (PKD) and unpolished (UKD) Kaab Dum rice extracts on the migration (wound closure) of TGF-β-induced HaCaT cells using a wound healing assay at 24 and 48 h.** A: Wound closure at 24 and 48 h. B: Percentage of wound healing at 24 h. C: Percentage of wound healing at 48 h. Statistical significance: * ($p < 0.05$), ** ($p < 0.01$), *** ($p < 0.001$), **** ($p < 0.0001$).

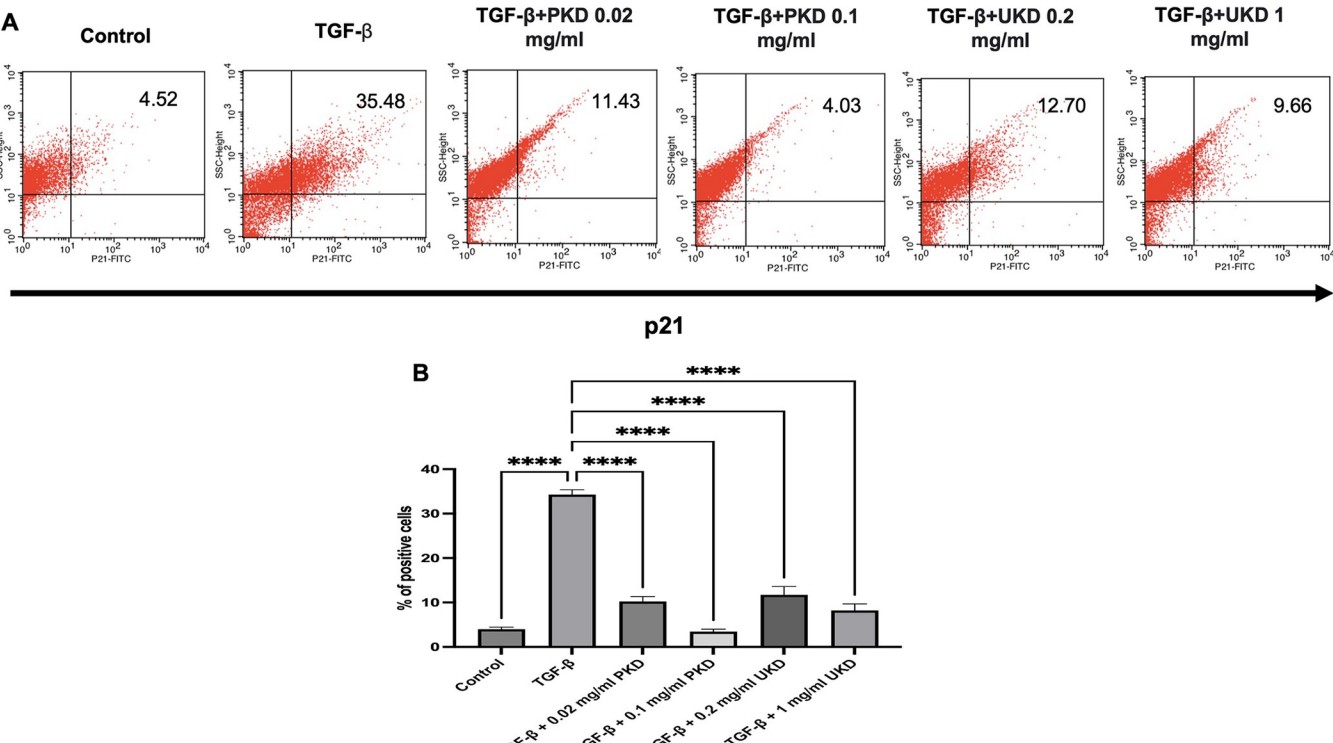

**Fig 6. Effect of polished (PKD) and unpolished (UKD) Kaab Dum rice extracts on p21 protein expression in TGF-β-induced HaCaT cells using flow cytometry at 24 h.** A: p21 protein expression. B: Percentage of p21 protein expression in positive cells. Significance level: **** ($p < 0.0001$).

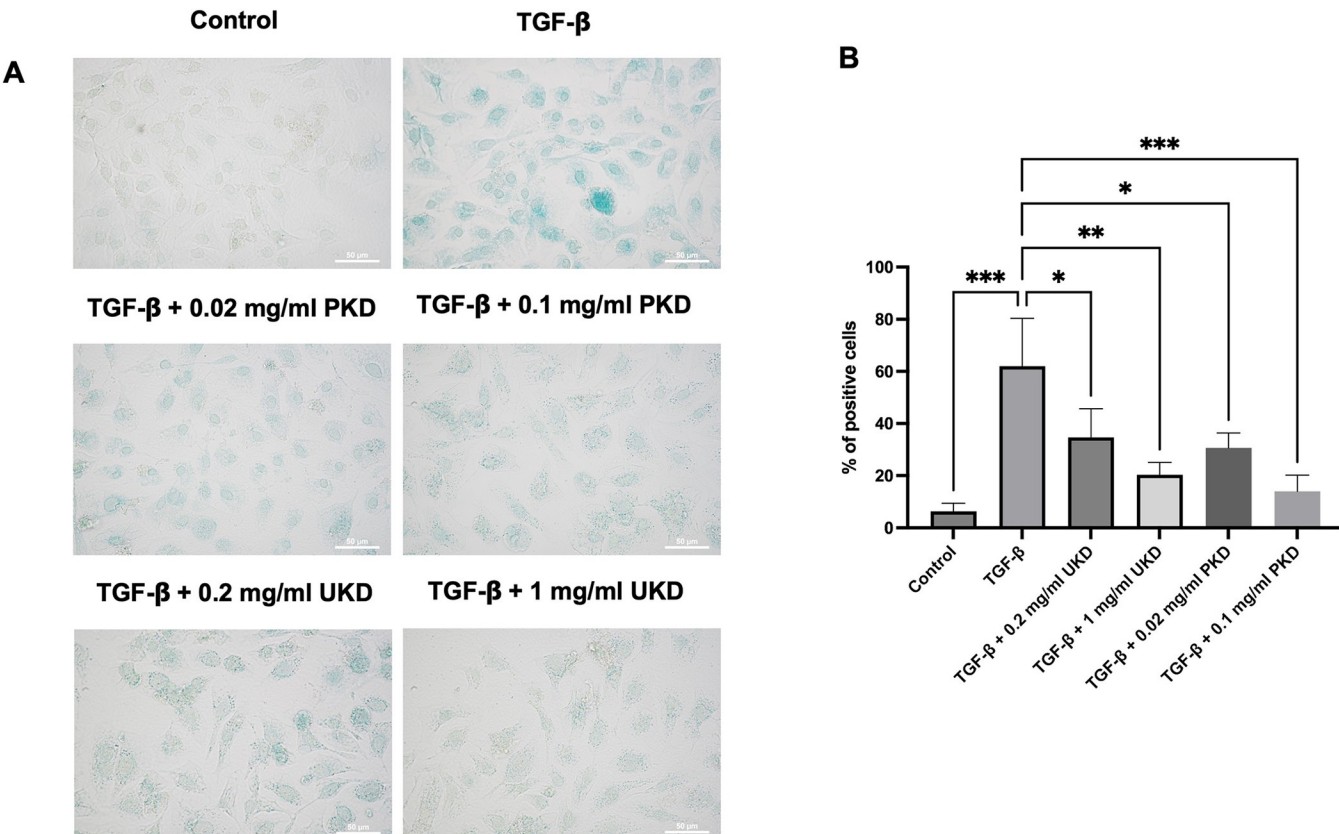

**Fig 7. Effects of polished (PKD) and unpolished UKD) Kaab Dum rice extracts on senescence of TGF-β-induced HaCaT cells using senescence-associated β-galactosidase (SA-β-gal) staining at 24.** A: Senescent cells stained blue with SA-β-gal enzyme reaction. B: Percentage of positive cells. Significance: * ($p < 0.05$), ** ($p < 0.01$), *** ($p < 0.001$).

## Discussion

Kaab Dum rice is a traditional local rice variety that has been cultivated for more than 50 years in the Pak Phanang Basin, Nakhon Si Thammarat province. It is cooked for consumption, and due to its firm texture when cooked, it is also used as a raw material for processing various products, especially as a key ingredient in making traditional local delicacies in the region. An analysis of rice compositions revealed that unpolished rice contains higher magnesium levels than polished rice, with Kaab Dum rice displaying particularly elevated magnesium content. Magnesium plays a pivotal role as a cofactor for enzymes involved in vital processes, including protein synthesis, muscle and nerve transmission, active transport of calcium and potassium ions across cell membranes, structural bone development and immunological functions like macrophage activation and lymphocyte proliferation [20]. In addition, Kaab Dum rice contains lower sodium levels. Sodium is crucial for regulating water and fluid balance, as well as controlling blood pressure. There is a significant connection between sodium intake and various health events, including strokes and coronary heart disease [21]. Furthermore, the analysis reveals that Kaab Dum rice is rich in zinc. Zinc deficiency can lead to impaired physical growth and development, impacting the epidermal, gastrointestinal, central nervous, immune, skeletal and reproductive systems [22].

Chemical analysis of Kaab Dum rice extract has identified a range of compounds, with sugars, lipids, an essential amino acid and choline as the major constituents. Tryptophan, an essential amino acid, plays a crucial role in protein synthesis, which is vital for various biological functions such as tissue growth, repair and maintenance. It also serves as a precursor for

serotonin, a neurotransmitter that regulates mood, sleep and appetite [23]. C16 Sphinganine, an essential component of sphingolipids, is involved in critical processes like cell growth, differentiation and apoptosis [24]. Glycerophospho-N-oleoyl ethanolamine, a naturally occurring lipid molecule, plays a role in various physiological processes, particularly in the regulation of appetite and energy balance [25, 26]. Another lipid molecule, linoleoyl ethanolamide, is also involved in appetite regulation, the endocannabinoid system and may have potential roles in obesity management [27].

Chronic wounds, characterized by their prolonged inability to heal within the expected time frame, pose a significant medical challenge [28]. TGF-β coordinates critical processes such as re-epithelialization, angiogenesis, fibroblast proliferation and extracellular matrix (ECM) production, leading to granulation tissue formation and wound contraction. In the remodeling phase, TGF-β influences the balance between collagen degradation by matrix metalloproteinases (MMPs) and the presence of endogenous tissue inhibitors of MMPs (TIMPs), whose expression it also regulates [11, 29, 30]. TGF-β is focusing in this study because the previous evidence showed TGF-β as the main factor producing chronic wounds, including diabetic ulcers [10] and pressure wounds [31]. Our study uses TGF-β to induce chronic wound conditions, where TGF-β also induces oxidative stress by stimulating the production of reactive oxygen species (ROS). Our research demonstrates that TGF-β induces oxidative stress in HaCaT cells, while the application of Kaab Dum rice extract effectively reduces cellular oxidant levels. This oxidative stress can subsequently lead to ER stress, resulting in the accumulation of misfolded or unfolded proteins, in accordance with previous studies [32, 33]. Protein kinase R-like endoplasmic reticulum kinase (PERK) is a key mediator of ER stress, playing a crucial role in the regulation of essential cellular functions [34]. However, when ER stress becomes irreversible, it can lead to the deterioration of cellular functions, often resulting in cell death [35]. Our study found that TGF-β-induced chronic wounds exhibited signs of ER stress, indicated by the presence of PERK. Furthermore, ER stress hindered the migration of HaCaT keratinocytes. However, the administration of Kaab Dum rice extract significantly improved ER stress and the migration of keratinocytes. We propose that the high magnesium and zinc content in the extract may contribute to the reduction of oxidative stress. Furthermore, the presence of tryptophan in Kaab Dum rice extract has the potential to enhance antioxidant status, alleviate inflammation and ER stress in TGF-β-induced HaCaT keratinocytes. These tryptophan-related effects are consistent with findings in a previous study [36].

The cyclin-dependent kinase inhibitors (CDKIs) play a crucial role in regulating cell cycle processes. P21, a CDKI, exerts anti-proliferative effects through a P53-dependent mechanism. It is activated by p16INK4A, leading to the inhibition of CDK2-cyclin E complex. This inhibition results in the hypo-phosphorylation of pRB, leading to cell cycle arrest [37]. Senescence is often identified by the presence of SA-β-gal activity, which is a reliable marker in cultured cells and mammalian tissues [12]. Our study shows that TGF-β induces senescence in HaCaT cells by upregulating p21 protein and SA-β-gal expression, while treatment with Kaab Dum rice extract reduces p21 protein and SA-β-gal expression. We attribute these effects to the antioxidant and/or anti-inflammatory properties of the nutritional and chemical components found in Kaab Dum rice extract, including linoleoyl ethanolamide, phosphatidylcholine, magnesium, zinc and tryptophan. Previous studies have indicated that both linoleoyl ethanolamide and phosphatidylcholine exert anti-inflammatory effects [38, 39].

## Conclusions

This study induced chronic wounds in HaCaT cells using TGF-β, which led to ER stress, as confirmed by the expression of the PERK protein. Subsequently, this ER stress caused cell

cycle arrest in HaCaT cells, evidenced by an increase in a CDKI, p21 protein. This process resulted in the cells entering a state of senescence, as confirmed by staining with the SA-β-gal. Of particular interest, the study revealed the effectiveness of extracts from Kaab Dum rice in promoting wound healing in the chronic wound model by reducing ER stress and senescent cells. Nevertheless, further research is necessary to gain a deeper understanding of the underlying mechanisms involved in this process. Moreover, this research holds the potential to pave the way for the development of natural products for the treatment of chronic wounds, addressing a significant medical concern.

## Supporting information

**S1 File.**
(XLSX)

## Acknowledgments

We are thankful to the President and all the staff of Walailak University and Mahidol University for their kind support.

## Author Contributions

**Conceptualization:** Witchuda Payuhakrit, Sarawoot Palipoch.

**Data curation:** Witchuda Payuhakrit, Ratsada Praphasawat, Sarawoot Palipoch.

**Formal analysis:** Witchuda Payuhakrit, Gorrawit Yusakul, Ratsada Praphasawat, Sarawoot Palipoch.

**Funding acquisition:** Sarawoot Palipoch.

**Investigation:** Witchuda Payuhakrit, Pimchanok Panpinyaporn, Wilunplus Khumsri, Gorrawit Yusakul, Nitra Nuengchamnong, Sarawoot Palipoch.

**Methodology:** Witchuda Payuhakrit, Pimchanok Panpinyaporn, Wilunplus Khumsri, Gorrawit Yusakul, Nitra Nuengchamnong, Sarawoot Palipoch.

**Project administration:** Sarawoot Palipoch.

**Resources:** Sarawoot Palipoch.

**Supervision:** Sarawoot Palipoch.

**Validation:** Witchuda Payuhakrit, Gorrawit Yusakul, Ratsada Praphasawat, Sarawoot Palipoch.

**Visualization:** Witchuda Payuhakrit, Ratsada Praphasawat, Sarawoot Palipoch.

**Writing – original draft:** Witchuda Payuhakrit, Ratsada Praphasawat, Sarawoot Palipoch.

**Writing – review & editing:** Sarawoot Palipoch.

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
