## [Decision Letter · Decision Letter 0]

9 Jan 2024

PONE-D-23-34554Enhancing chronic wound healing with Thai indigenous rice variety, Kaab Dum: exploring ER stress and senescence inhibition in HaCaT keratinocyte cell linePLOS ONE

Dear Dr. Palipoch,

Thank you for submitting your manuscript to PLOS ONE. After careful consideration, we feel that it has merit but does not fully meet PLOS ONE’s publication criteria as it currently stands. Therefore, we invite you to submit a revised version of the manuscript that addresses the points raised during the review process.

We look forward to receiving your revised manuscript.

Kind regards,

Gopinath Ponnusamy Manogaran, PhD

Academic Editor

PLOS ONE

“the Office of the Royal Development Projects Board, Thailand.”

Reviewers' comments:

Reviewer's Responses to Questions

**Comments to the Author**

1. Is the manuscript technically sound, and do the data support the conclusions?

Reviewer #1: Yes

Reviewer #2: Partly

Reviewer #3: Yes

2. Has the statistical analysis been performed appropriately and rigorously? 

Reviewer #1: Yes

Reviewer #2: Yes

Reviewer #3: Yes

3. Have the authors made all data underlying the findings in their manuscript fully available?

Reviewer #1: Yes

Reviewer #2: Yes

Reviewer #3: Yes

4. Is the manuscript presented in an intelligible fashion and written in standard English?

Reviewer #1: Yes

Reviewer #2: Yes

Reviewer #3: Yes

5. Review Comments to the Author

Reviewer #1: Dear Prof Dr/Editor,

Greetings!

Thank you for your invitation to review the manuscript entitled ‘’ Enhancing chronic wound healing with Thai indigenous rice variety, Kaab Dum: exploring ER stress and senescence inhibition in HaCaT keratinocyte cell line’’.

Please, see below my comments:

Introduction:

1. Please, explain the relationship between senescence, inflammation, and delayed wound healing.

Materials and methods

2. Please, mention the ethanol: water (V/V) used for your extract.

3. Please, mention the final concentration of your stock in DMSO.

4. Please, mention the solute for MTT solution. Is it medium or PBS?

5. The concentration of MTT is 0.5 mg per L or ml???

6. What is the concentration of DMSO after addition of your extract to the medium???

7. What is the type of fixative you used in SA-β-gal staining assay?

8. Correct the title of Table 1 ………polished and unpolished……..

Results

9. Please, revise the abbreviation of TGF-β in all result section.

10. In the wound healing assay, the image 3 and 4 of 0 h is not similar in size as others.

Discussion

11. The discussion section containing too long paragraphs.

Reviewer #2: This article presents a study on the therapeutic potential of Kaab Dum rice extract in the context of chronic wound healing. The investigation focused on the effects of the indigenous rice variety on TGF-β-induced HaCaT cells, specifically examining oxidative stress, endoplasmic reticulum (ER) stress, cell cycle arrest, and cellular senescence. The findings suggest a reduction of oxidative stress, inhibition of ER stress, and promotion of wound healing. However, there are areas where the study could be strengthened:

1. A more detailed investigation into the mechanisms by which Kaab Dum rice extract affects wound healing is necessary to fully understand its therapeutic potential.

2. The absence of in vivo studies or animal models is a significant limitation. Such studies are crucial for assessing the physiological relevance of the rice extract in a more complex biological system.

3. Comparing the effectiveness of Kaab Dum rice extract with other treatments for chronic wounds would provide context and further insight.

4. Detailed information about the reagents used in the study should be provided, including the source, stock number, tradename, company, city, and country. This information is critical for the reproducibility of the research.

5. An inconsistency in Figure 5 regarding the cell scratching assay should be addressed to ensure the validity of the results. The scratch area at 0 hr for the groups treated with TGF-β+0.02 mg/ml PKD and TGF-β+0.1 mg/ml PKD does not match the initial scratch area of the other groups.

6. What is the authors' rationale for choosing TGF-β induced HaCaT cells as a model of chronic wound? What type of chronic wound is this? Further justification and discussion are needed.

Reviewer #3: Dear Authors,

Your manuscript is well-structured and informative. Addressing the following comments will enhance your study's clarity and scientific rigor.

1. Antibiotic-Antimycotic Solution: For clarity, kindly mention the composition of the antibiotic-antimycotic solution used in your experiments.

2. MTT Solution Concentration: Clarify whether the MTT solution concentration of 0.5 mg/L should instead be reported as 0.5 mg/mL for accuracy.

3. Transforming Growth Factor-β (TGF-β): Provide the source and product number of the TGF-β used in your study for better reproducibility.

4. Rice Extract Concentrations: Explain the rationale behind using different concentrations for polished and unpolished Kaab Dum rice extracts (0.02 and 0.1 mg/mL for polished; 0.2 and 1 mg/mL for unpolished). What factors influenced this selection?

5. Scratch Wound as Chronic Wound: Justify why the scratch wound model was chosen to represent a chronic wound in your study.

6. % of Control and % of Inhibition: Clarify the meaning of "% of control." Does it refer to "% of inhibition"? Please provide clarity on these percentages for better understanding.

6. PLOS authors have the option to publish the peer review history of their article (what does this mean?). If published, this will include your full peer review and any attached files.

Reviewer #1: No

Reviewer #2: **Yes: **Xuqiang Nie

Reviewer #3: No

---

## [Author Response · Author response to Decision Letter 0]

25 Jan 2024

Dear Editor

We edited and added information according to the editor's suggestions as follows:

1. Our manuscript meets PLOS ONE's style requirements.

2. We state: "The funders had no role in study design, data collection and analysis, decision to publish, or preparation of the manuscript." in our cover letter.

3. We provide a complete Data Availability Statement in the submission form, we write "All data are in the manuscript".

4. I have an ORCID iD and that it is validated in Editorial Manager. 

We made changes based on reviewers' comments. Details are in the attached file “response to reviewers”.

Yours sincerely,

Assoc.Prof.Dr. Sarawoot Palipoch

Corresponding Author

---

## [Decision Letter · Decision Letter 1]

6 Mar 2024

PONE-D-23-34554R1Enhancing chronic wound healing with Thai indigenous rice variety, Kaab Dum: exploring ER stress and senescence inhibition in HaCaT keratinocyte cell linePLOS ONE

Dear Dr. Palipoch,

Thank you for submitting your manuscript to PLOS ONE. After careful consideration, we feel that it has merit but does not fully meet PLOS ONE’s publication criteria as it currently stands. Therefore, we invite you to submit a revised version of the manuscript that addresses the points raised during the review process.

We look forward to receiving your revised manuscript.

Kind regards,

Gopinath Ponnusamy Manogaran, PhD

Academic Editor

PLOS ONE

Journal Requirements:

Reviewers' comments:

Reviewer's Responses to Questions

**Comments to the Author**

1. If the authors have adequately addressed your comments raised in a previous round of review and you feel that this manuscript is now acceptable for publication, you may indicate that here to bypass the “Comments to the Author” section, enter your conflict of interest statement in the “Confidential to Editor” section, and submit your "Accept" recommendation.

Reviewer #2: All comments have been addressed

Reviewer #3: All comments have been addressed

2. Is the manuscript technically sound, and do the data support the conclusions?

Reviewer #2: Partly

Reviewer #3: Yes

3. Has the statistical analysis been performed appropriately and rigorously? 

Reviewer #2: Yes

Reviewer #3: Yes

4. Have the authors made all data underlying the findings in their manuscript fully available?

Reviewer #2: Yes

Reviewer #3: Yes

5. Is the manuscript presented in an intelligible fashion and written in standard English?

Reviewer #2: Yes

Reviewer #3: Yes

6. Review Comments to the Author

Reviewer #2: I appreciate the authors' responses and I have only a couple of residual questions.

1. The LC-ESI-QTOF-MS/MS chromatogram should be marked with the main peak information.

2. A few typographical errors also remain (i.e. the word " TGF-β" on pg 20-21, Table 2, line 4, [M+Na}? ).

Reviewer #3: (No Response)

7. PLOS authors have the option to publish the peer review history of their article (what does this mean?). If published, this will include your full peer review and any attached files.

Reviewer #2: **Yes: **Xuqiang Nie

Reviewer #3: No

---

## [Author Response · Author response to Decision Letter 1]

7 Mar 2024

Dear Editor and Reviewers

 We made changes based on reviewers' comments. Details are in the attached file “response to reviewers”.

Yours sincerely,

Assoc.Prof.Dr. Sarawoot Palipoch

Corresponding Author

---

## [Editor Report · Decision Letter 2]

10 Apr 2024

Enhancing chronic wound healing with Thai indigenous rice variety, Kaab Dum: exploring ER stress and senescence inhibition in HaCaT keratinocyte cell line

PONE-D-23-34554R2

Dear Dr. Palipoch,

We’re pleased to inform you that your manuscript has been judged scientifically suitable for publication and will be formally accepted for publication once it meets all outstanding technical requirements.

Kind regards,

Gopinath Ponnusamy Manogaran, PhD

Academic Editor

PLOS ONE